# Effect of Selenium and Selenoproteins on Radiation Resistance

**DOI:** 10.3390/nu16172902

**Published:** 2024-08-30

**Authors:** Shidi Zhang, Guowei Zhang, Pengjie Wang, Lianshun Wang, Bing Fang, Jiaqiang Huang

**Affiliations:** 1Beijing Advanced Innovation Center for Food Nutrition and Human Health, Department of Nutrition and Health, China Agricultural University, Beijing 100083, China; sy20233313708@cau.edu.cn (S.Z.); wpj1019@cau.edu.cn (P.W.); 2Limited Liability Company of Hongda Salt Industry, Hoboksar Mongol Autonomous County, Tacheng 834700, China; 3College of Fisheries and Life Science, Dalian Ocean University, Dalian 116023, China

**Keywords:** selenium, radiation resistance, mechanism

## Abstract

With the advancement of radiological medicine and nuclear industry technology, radiation is increasingly used to diagnose human health disorders. However, large-scale nuclear leakage has heightened concerns about the impact on human organs and tissues. Selenium is an essential trace element that functions in the body mainly in the form of selenoproteins. Selenium and selenoproteins can protect against radiation by stimulating antioxidant actions, DNA repair functions, and immune enhancement. While studies on reducing radiation through antioxidants have been conducted for many years, the underlying mechanisms of selenium and selenoproteins as significant antioxidants in radiation damage mitigation remain incompletely understood. Therefore, this paper aims to provide new insights into developing safe and effective radiation protection agents by summarizing the anti-radiation mechanism of selenium and selenoproteins.

## 1. Introduction

In recent years, radiation has been widely used in a variety of disciplines, including medical imaging diagnosis, radiotherapy, and radiation sterilization, thereby facilitating the advancement of the medical and food industries. While promoting industrial development, radiation also has some negative effects that cannot be ignored, such as nuclear leakage, medical radiation exposure, and radiation from electronic products. Recently, Japan’s nuclear sewage discharge events have increased the risk of radiation to human life, leading to more concerns. Selenium and selenoproteins have attracted the interest of researchers due to their potential in radiation resistance. Studying the effects and mechanisms of selenium and selenoproteins on radiation resistance will help researchers find safer anti-radiation methods and develop effective anti-radiation drugs.

Radiation is categorized into ionizing radiation and non-ionizing radiation. This paper focuses on the damage that ionizing radiation causes to human organs and tissues. Ionizing radiation generates electromagnetic waves of various ions, leading to the production of harmful free radicals in the body, which can result in decreased immunity, skin redness, cancer, and other adverse effects that can cause serious harm and even death [1]. The damage mechanisms can be classified into three types: direct action, indirect action, and bystander effect. Direct damage occurs when high-energy ionizing radiation directly affects important macromolecules with biological activity, such as nuclear DNA, proteins or lipid membranes. The key target of direct damage is DNA, which can result in DNA single-strand and double-strand breaks [2]. Indirect damage is the primary form of ionizing radiation’s impact. Ray energy acts on water molecules within cells, causing the ionization of water molecules, producing various reactive oxygen species (ROS), resulting in oxidative stress [3,4]. Under oxidative stress conditions, excessive ROS can cause DNA damage and inhibit DNA repair. Oligosaccharide molecules in the cell membrane are oxidized to unsaturated free radical dimers, which impair the cell membrane function and cause protein dysfunction. Sustained injury response can affect related signal transduction, such as activation of the p53 signaling pathway, thereby stimulating the expression of cyclin-dependent kinase inhibitors p21 and p16, causing cell senescence [5]. Mitochondrial damage causes a large amount of ROS to escape, induces the loss of mitochondrial respiratory chain complex activity, reduces the efficiency of oxidative phosphorylation level, causes cell apoptosis, leads to mitochondrial cascade damage, and results in the doubling of free radical level in the whole cell, which continues to cause more serious damage [6]. The bystander effect refers to transmitting signals to adjacent or non-irradiated distal cells, causing them to be affected by radiation and exhibit various biological dysfunctions, including DNA damage, chromosomal aberrations, and micronuclei formation [7]. Furthermore, DNA damage and continuous oxidative stress can destabilize growth and secretion in bystander cells. Consequently, the effects induced in bystander cells may even spread to secondary observers and their offspring. Regarding radiation damage mechanisms, existing anti-radiation measures mainly focus on oxidation control, DNA repair enhancement, immune system strengthening, etc. While radio-protective agents appear effective at reducing radiation-induced damage, most synthetic agents have toxic side effects that make them impractical for long-term application. Therefore, searching for new radiation protective measures is very necessary.

Selenium is an essential trace element in the human body and is available in inorganic and organic forms. Inorganic selenium compounds include selenite, elemental selenium, and selenium acids; organic selenium includes selenoproteins, selenomethionine (SeMet), selenium yeast, and selenium polysaccharide. Selenium deficiency in the human body may lead to various diseases [8,9], such as Keshan disease and carbon metabolism disorders, or a decrease in resistance to certain viruses [10,11], and increase in radiation-induced micronucleus formation [12]. Selenium plays various biological functions in the form of selenoproteins in the human body, with 25 distinct species of selenium protein genes identified [13]. Selenoproteins are intracellular antioxidants that play an important role in limiting oxidative damage by acting as oxidoreductases for glutathione peroxidases (GPXs), thioredoxin reductases (TRXRDs), and iodothyronine deiodinases (DIOs). Additionally, several selenoproteins, such as selenoprotein P (SELENOP), selenoprotein H (SELENOH), selenoprotein M, selenoprotein T (SELENOT), and selenoprotein W (SELENOW), possess selenocysteine-like structural domains, suggesting potential redox-related functions [14]. Selenium and selenoproteins have various biological functions. Numerous studies have indicated that selenium exhibits promising anti-tumor effects through its capacity of inhibiting tumor cell development, inducing tumor cells to die, and scavenging reactive oxygen species via selenoproteins [15]. Furthermore, selenium and selenoproteins can enhance immune system regulation [16], heavy metal resistance [17], cardiovascular system protection [18], and radiation resistance. Therefore, selenium and selenoproteins hold promise as potential radio-protective agents.

Selenium and selenoproteins play a crucial role in eliminating free radicals and repairing DNA, thereby contributing to immune regulation and protection against radiation damage. Despite this, there is a lack of comprehensive literature on the anti-radiation properties of selenium and selenoproteins, as well as their underlying mechanisms. Therefore, this review aims to elucidate the anti-radiation effects and mechanisms of selenium and selenoproteins. It also outlines potential prospective future research avenues for their application in anti-radiation therapy.

## 2. Selenoproteins and Their Anti-Radiation Effects

Ionizing radiation has the potential to cause damage to tissues and may induce cell carcinogenesis. Studies have indicated that selenium and selenoproteins may play a significant role in various stages of radiation-induced damage. Certain selenium compounds can act preventatively during the initial stage of radiation exposure, while others are involved in the regulation, repair, and treatment of tissue damage following exposure to radiation [19].

The following section provides descriptions of selenoproteins with anti-radiation activity, as well as their corresponding effects in mitigating radiation (Table 1).

### 2.1. Glutathione Peroxidases (GPXs)

The main biological effects of selenium in the human body are mediated by selenoproteins. The glutathione peroxidase (GPX) group is an important component of the selenium protein family, serving as a crucial antioxidant and scavenger of ROS in the body. Furthermore, it regulates the REDOX balance in tissues [20]. Currently, eight GPX isoforms have been found within the GPX family, namely GPX1–GPX8. Among them, GPX1–GPX4 and GPX6 have SeCys as a catalytic site. Thus, they are known as SeCys-containing GPXs. Research has indicated that the molecular functions of GPXs include antioxidant activity, peroxidase activity, and glutathione peroxidase activity. Additionally, GPXS are implicated in glutathione metabolism, arachidonic acid metabolism, and other physiological metabolic processes [21]. Both GPX1 and GPX4 are widely expressed in various tissues and may play a pivotal role in the anti-radiation process. The subsequent content will primarily elaborate the potential anti-radiation pathways of GPX1 and GPX4.

#### 2.1.1. Glutathione Peroxidase 1 (GPX1)

GPX1 is a selenoprotein with an active site containing selenocysteine. It is expressed universally in all tissues and shows high expression levels in red blood cells, liver, lungs, and kidneys. Furthermore, it can be found in cytosol and mitochondria [22]. It plays a crucial role in the cellular antioxidant defense mechanism, and can also promote DNA repair, regulate cardiovascular function, and exhibit anti-cancer activity [23]. Because it is involved in so many biological processes, the essential selenoprotein GPX1 is vital for radiation defense.

GPX1 plays a crucial role in removing the excess free radicals generated by radiation, regulating the REDOX balance in the body, and reducing cell damage caused by radiation, thereby enhancing radiation resistance. Research reports indicate that ROS directly contribute to oxidative stress accumulation and trigger additional ROS production through various mechanisms, further disrupting the body’s REDOX balance. According to Diane E. Handy and colleagues, the accumulation of ROS may stimulate REDOX-mediated signal transduction events, including activation of transcription factors such as nuclear factor kappa B (NF-κB), which predominantly occurs in GPx1-deficient endothelial cells and leads to inflammation activation [24]. GPX1 can catalyze the reaction between glutathione (GSH) and some superoxides, to regulate the accumulation of these superoxides. This adjustment helps regulate the generation and accumulation of other forms of ROS, ultimately achieving antioxidant effects [25]. GPX1 also promotes the repair of radiation-induced DNA damage. A study conducted by Morais investigated the changes in selenium levels after exposure to ionizing radiation and H2AX phosphorylation. H2AX is a histone that recruits DNA damage repair proteins to injury sites, and its phosphorylation level indicates the extent of DNA damage repair through its phosphorylation level. The study demonstrated that GPX1 contributes to DNA damage repair [26]. Radiation has the potential to induce tumors and cell carcinogenesis. GPX1 can exert an anticancer effect through various mechanisms. Firstly, it can directly impact specific tumor suppressor genes in cancer cells, such as activating the p53 gene and promoting cancer cell apoptosis. Secondly, it can enhance the sensitivity of tumor cells to chemotherapy drugs [27]. Thirdly, it can inhibit tumorigenesis signaling pathways through antioxidant mechanisms and other means. For example, promoting Nrf2 signaling pathway activation is implicated in tumor prevention, which is considered a primary mechanism of action for many chemopreventive agents in inhibiting tumor formation by preventing ROS-mediated DNA damage [28,29].

#### 2.1.2. Glutathione Peroxidase 4 (GPX4)

Several selenium glutathione peroxidases have been identified to contribute to radiation resistance, with GPX1 and GPX4 being particularly important. Radiation exposure can trigger oxidative stress, leading to potentially severe metabolic dysfunctions such as membrane lipid peroxidation and DNA damage [30]. While GPX4 effectively inhibits lipid peroxide, it also promotes DNA repair to alleviate radiation-induced damage and adverse effects, thereby maximizing its role in radiation therapy.

GPX4 has a more significant function in radiation therapy because, in addition to its efficient inhibition of lipid peroxidation, it also helps repair DNA damage in order to reduce radiation-induced damage and side effects [31]. Within the DNA structure, thymine residues possess the highest electron affinity and are, therefore, the most susceptible sites for free radical injury. GPX4 can reduce thymine peroxide caused by ionizing radiation, reducing DNA damage [32]. Overexpression of GPX4 can totally suppress interleukin-1-induced NF-κB activation, while GPX1 exerted only moderate effects on this pathway [32]. Furthermore, the experimental results demonstrate that GPX4 effectively regulates many apoptosis markers in various cells, such as the release of cytochrome c, DNA fragmentation, and inhibition of NF-κB. For example, it protects the preferred substrate of cytochrome c, cardiolipin, from peroxidation and inhibits the release of cytochrome c from mitochondria in response to oxidative stress exposure, effectively regulating apoptosis [33,34].

### 2.2. Selenoprotein P (SELENOP)

SELENOP is an extracellular glycoprotein, a selenium transporter, and a major supplier of selenium in tissues [35]. It is widely distributed in almost all tissues and it contains ten SeMets, of which the *N*-terminal SeMet have antioxidant functions, while the *C*-terminal nine SeMets are the main suppliers of selenium to tissues. SELENOP also plays a crucial role in certain aspects of immune function. The REDOX tone also influences macrophage differentiation, and several studies have indicated that SELENOP is especially important in this process [36]. Studies suggest that SELENOP is a previously unidentified antioxidant protein capable of regulating the late accumulation of ROS and mitigating toxicity in normal human fibroblasts following radiation exposure. The expression of SELENOP effectively suppresses radiation-induced late ROS accumulation and associated toxicity [37,38]. This indicates that SELENOP is a crucial regulator of the cellular response to radiation. Furthermore, the antioxidant function of SELENOP suggests a possible involvement in cancer prevention, particularly in cases characterized by heightened oxidative stress, such as inflammatory bowel cancer. Research conducted by Caitlyn W. Barrett et al. has demonstrated that SELENOP acts as a haploid tumor suppressor in inflammatory carcinogenesis [39]. At the same time, SELENOP also influences macrophage polarization, reducing the incidence of inflammatory tumors by attenuating the activation of pro-inflammatory cells [39,40,41].

### 2.3. Selenoprotein S (SELENOS)

Selenoprotein S (SELENOS) is a small, inherently disordered membrane protein associated with various cellular functions, including inflammatory processes and cellular stress responses. It is most notably recognized for its involvement in the endoplasmic reticulum-associated degradation (ERAD) pathway [42]. This pathway involves the extraction of misfolded proteins or misassembled protein complexes from the endoplasmic reticulum (ER) into the cytosol for proteasomal degradation. Radiation can lead to the accumulation of misfolded proteins in the ER, causing ER stress and triggering the unfolded protein response (UPR), further promoting inflammation. The participation of ERAD SELENOS is crucial in the UPR process, as it regulates the removal of misfolded proteins or incorrectly assembled protein complexes from the ER to the cytoplasm, resulting in proteasomal degradation and alleviating radiation-induced endoplasmic reticulum stress. It is evident that SELENOS is also important in responding to radiation [43].

### 2.4. Selenoprotein I (SELENOI)

The selenoprotein I (SELENOI) group is a unique among selenoproteins. It acts as an ethanolamine phosphate transferase (Ept) rather than a REDOX catalyst. This enzyme is critical for synthesizing phosphatidyl ethanolamine (PE) and alkenyl PE in pluripotent stem cells, as well as the proliferation of tumor cells [44]. Its involvement in metabolic reprogramming is particularly significant [45]. Selenoprotein I may play crucial functions in T cell activation and metabolic reprogramming. As a component, SELENOI can maintain ethanolamine phospholipid synthesis in cell metabolic reprogramming, thus contributing to the balance of cell metabolism and playing an essential function in the process of T cell activation. Therefore, SELENOI can be instrumental in regulating immune cells to combat radiation-induced immune responses and exhibits radiation resistance [46].

In conclusion, multiple selenoproteins can exert anti-radiation effects through different mechanisms (Figure 1). Glutathione peroxidase 1 (GPX1) and glutathione peroxidase 4 (GPX4) contribute to anti-radiation activities by enhancing DNA repair and regulating anti-inflammatory and anticancer signaling pathways. GPX4 is especially effective at removing lipid peroxidation in response to radiation-induced oxidative stress. GPX1 and SELENOP function by scavenging excess free radicals. GPX4 and SELENOI can also promote radiation-induced apoptosis through immune regulation and other pathways. Meanwhile, SELENOS primarily exerts its anti-radiation activity by relieving endoplasmic reticulum stress.

## 3. Radiation Resistance Mechanism of Selenium and Selenoproteins

Selenium and selenoproteins exhibit the potential to mitigate the effects of radiation, a capability closely linked to biological activities such as anti-oxidation, immune regulation, and apoptosis control. Research has demonstrated that selenium and selenoproteins combat radiation by reducing oxidative damage, regulating the immune system, managing cell apoptosis, and protecting DNA and the hematological system.

### 3.1. Improves Oxidative Damage

The REDOX system dynamically regulates the production of active oxygen free radicals under normal physiological conditions to maintain a balanced level. Oxidative stress occurs when there is an imbalance between oxidation and anti-oxidation in the body, leading to a pathological condition characterized by high ROS levels [47]. Radiation stimulation causes the ionization and excitation of water molecules, resulting in the generation of an excessive amount of reactive oxygen species. These ROS will subsequently cause a free radical chain reaction, causing oxidative stress. Consequently, excess ROS destroy biological macromolecules such as DNA, lipids, proteins, and mitochondria [3]. Damage to nucleic acids, lipid peroxidation, and protein damage might cause alterations in essential antioxidant enzymes such as SOD and GSH-Px [48]. Additionally, non-enzymatic antioxidants play a crucial role in combating free radicals. These substances include small molecular compounds with antioxidant effects, such as GSH synthesized in cells, ceruloplasmin, and vitamin E [49].

It has been reported that selenium and selenoproteins can scavenge ROS directly. In biological systems, selenium compounds tend to be restored; it is a good nucleophilic reagent. They are used in heavy metal detoxification to produce ROS-required metal ion complexation, which indirectly removes ROS. Selenium also acts as a cofactor for various antioxidant enzymes, such as GSH-Px, and contributes to antioxidant effects [50]. Sodium selenite is the first selenium compound to be studied for radiation protection in a mouse model [51]: it has been shown to be efficient when combined with vitamin E prior to gamma irradiation in mice, preventing the radiation-induced decrease of antioxidant enzymes [52]. Supplementation with sodium selenite has been shown to increase serum GPX activity while decreasing treatment-induced oxidative stress. 3′-3′ Diselendipropionic acid (DSePA) has been identified as a potent radio-protectant. It has shown efficacy as a free radical scavenger, GPX mimic, and anti-hemolytic agent. The high levels of ROS observed in GPX1 knockout mice during the injury response highlight the importance of antioxidant pathways in limiting ROS-induced cell damage [53]. Furthermore, numerous studies have demonstrated that GPX4 regulates lipid peroxidation and cell death induced by radiation. In summary, radiation can induce oxidative stress and pathological changes in the body. Improving oxidative damage is an important mechanism of selenium and selenoproteins against radiation damage.

### 3.2. Protecting DNA

Thus far, the findings indicate that safeguarding DNA is crucial for selenium and selenium proteins to resist radiation [3]. Ionizing radiation can degrade DNA molecules and damage nucleotides and their components. To overcome this damage, eukaryotes have a DNA repair system in place, which includes processes such as homologous recombination (HR), non-homologous end joining (NHEJ), base excision repair (BER), nucleotide excision repair (NER), and other mechanisms that contribute to DNA repair, genome stability, and aging. Double-stranded DNA breaks are repaired through HR or NHEJ, while single-strand breaks are repaired through BER or NER [54]. Chromosome DNA and protein complexes are sensitive to ionizing radiation, which can stimulate the ionization of cells in inland waters, resulting in a large quantity of reactive oxygen species. The strong oxidizing power of these species can destroy cell DNA, causing double-strand and single-strand breaks that result in chromosome aberrations. A series of in vitro studies, animal models, and human studies have demonstrated that selenium and selenium proteins can prevent DNA damage and enhance the activity of repair enzymes (such as DNA glycosylase, p53, BRCA1, Gadd45) providing protection [55]. GPX1 was shown to stimulate the expression of Gadd45, which is involved in DNA replication and damage repair [56]. SeMet protects against DNA damage by the creation of protein complexes with Ref1, p53, and Brca1. SeMet first induces the DNA repair branch of the p53 pathway. Then, Brca1 and Ref1 simultaneously interact with p53 to form protein complexes that target the induction of DNA repair responses [57].

### 3.3. Regulating Cell Apoptosis

As a strong stimulus, ionizing radiation can induce cell apoptosis through various mechanisms, including DNA damage, endoplasmic reticulum stress (ER stress), and ROS overload. However, selenium and selenoproteins can exert radio-protective effects by inhibiting cell apoptosis. Research has shown that different forms of selenium have different effects on cell apoptosis, with appropriate doses of selenium nanoparticles being more effective than sodium selenite and yeast selenium [58]. In addition, radiation-induced mitochondrial membrane damage can trigger the ER signaling pathway, ultimately leading to cell apoptosis [59]. Selenoprotein S is a transmembrane protein located in the endoplasmic reticulum (ER) and plasma membrane. It essential in the ER-associated degradation (ERAD) pathway and can regulate ER stress-induced cell apoptosis [60]. Furthermore, studies have indicated that GPX4 inhibits cell apoptosis in vivo through an endogenous apoptotic pathway [33]. This regulatory function is achieved by protecting the preferred substrate, cardiolipin, from oxidation and inhibiting cytochrome c release from mitochondria when exposed to oxidative stress [61,62].

### 3.4. Regulating Immune System

The immune system, which is made up of immune organs (such as the spleen, thymus, bone marrow, and lymph nodes), immune cells (including lymphocytes, macrophages, and granulocytes), and immune molecules (such as TNF, IFN, and IL), is responsible for preventing pathogens from invading the body and maintaining a healthy body [63]. Studies have shown that selenoproteins, such as selenoprotein I, play a significant role in the activation of T cells [45], while selenoprotein S regulates the steady-state aspects of immunity. This suggests that selenoproteins profoundly affect the activation and regulation of immune cells [64].

#### 3.4.1. MAPK Signaling Pathways and PI3K/AKT Signaling Pathway

Mitogen-activated protein kinase (MAPK) family, which includes extracellular signal-regulated kinase (ERK), p38 mitogen-activated protein kinase, and c-Jun *N*-terminal kinase (JNK). These three essential members participate in cell growth, proliferation, apoptosis, differentiation, and other growth processes [65]. Ionizing radiation stimulates the JNK and P38 pathways, activating classical MAPK signaling pathways and participating in immune regulation [66]. Some studies have shown that selenium deficiency downregulates the expression of 19 selenoproteins, increasing intracellular ROS levels. This, in turn, results in elevated levels of P38 and JNK gene expression, and increased expression of proteins involved in the mitochondrial apoptotic pathway. These findings indicate a close association between selenoproteins and ROS/MAPK-induced apoptosis, particularly necrotic apoptosis. Selenium and selenoproteins can regulate this pathway through multiple targets [67].

The PI3K/AKT pathway is a multifunctional signaling pathway related to defense, cell proliferation, and apoptosis. It regulates mammalian cell development, proliferation, and metabolic regulation. Activation of the PI3K/AKT pathway can accelerate DNA damage repair and inhibit ionizing radiation [68]. Experiments have revealed that hydrogel loaded with selenium nanoparticles (SeNPs) may eliminate reactive oxygen species and modify pH, which may protect cells from oxidative stress. Furthermore, these SeNPs induced macrophage M2 polarization through the PI3K/AKT and MAPK pathways, leading to reduced inflammatory cytokine production—a promising approach for mitigating radiation damage [69].

#### 3.4.2. Inflammation Signaling Pathway

As previously mentioned, exposure to ionizing radiation can lead to extensive oxidative damage, DNA breakage, and cell death. This results in the release of pro-inflammatory cytokines, including IL-1, IL-2, IL-4, IL-6, and TNF-α. Chronic upregulation of these cytokines can result in sustained generation of free radicals and mediate the occurrence of inflammation [70]. Inflammatory cytokines recruit immune cells and regulate the cellular microenvironment. Selenium and selenoproteins also regulate inflammatory factors and signaling pathways as part of their radio-protective mechanisms.

## 4. The Application of Selenium and Selenoproteins in the Treatment of Radiation-Related Diseases

Exposure to ionizing radiation can result in tissue damage in living organisms. The extent of the damage depends on the radiation exposure level. Acute clinical manifestations can impact the gastrointestinal tract, liver, kidneys, and lungs. Furthermore, organ and tissue damage may also occur. In addition to these organ systems, long-term observations have shown that radiation can also lead to damage to the kidneys, liver, and cardiovascular system, and even result in death [71]. As previously noted, selenium and selenium protein show significant potential in mitigating radiation damage. The selenium compounds used for radioprotection primarily consist of inorganic selenium compounds, selenoamino acids, and selenoproteins. This article will review some of the uses of selenium and selenium protein as radiation protective agents to alleviate multiple organ damage caused by radiation or treat related diseases.

### 4.1. Lung Injury and Treatment

The lungs are one of the most sensitive organs to ionizing radiation. Aside from cancer radiotherapy, the human body may be exposed to radiation environments for extended periods of time as a result of nuclear or radiation disasters. Lung injury is one of the most dangerous consequences of radiation exposure, which may lead to pneumonia and fibrosis. Severe cases may cause lung cancer and even threaten life [72]. Seleno-L-methionine is an effective antioxidant; previous research demonstrates its enormous potential in protecting and alleviating radiation damage [73]. We can apply seleno-L-methionine to prevent radiation-induced lung toxicity. It can reduce the accumulation of ROS and alleviate radiation-induced DNA damage and cell death by upregulating antioxidant enzymes to eliminate excessive ROS and directly chelating ROS. Amini et al. found that treating rats with seleno-L-methionine reduced the expression of these cytokines, hence reducing macrophage and lymphocyte infiltration [73]. 3′-3′ dithiopropionic acid (DSePA) is a simple, stable, and water-soluble organic selenium compound that can eliminate ROS and exhibits GPX-like activity. DSePA has been found to reduce oxidative damage to tissues by increasing the tissue levels of antioxidant defense enzymes, including GPX, thereby preventing mortality in a whole-body exposed mouse model. Kunwar et al.’s study demonstrated that DSePA treatment can decrease excessive free radicals and lipid peroxidation. This intervention also reduced the severity of pneumonia in mice. It affected the recruitment of lung radiation-induced inflammatory cells, such as pulmonary macrophages and neutrophils, suggesting that DSePA influences the inflammatory cell recruitment involved in the formation of radiation-induced lung injury [74]. SeNPs can prolong the radiation protection time without obvious side effects in the lung. Moreover, they exhibit potent cytotoxicity in cancer cells, but relatively little cytotoxicity in normal healthy cells [75].

### 4.2. Hematopoietic System

Bone marrow is one of the tissues most sensitive to radiation, and bone damage is a significant cause of death after high-dose irradiation. Studies have shown that prolonged radiation exposure may result in hematopoietic dysfunction, with increased levels of ROS and sustained oxidative stress potentially leading to long-term damage to radiation-induced hematopoietic stem cells (HSC). In an in vivo experiment with sodium selenite in mice, Jian Zhang et al. found that selenium can reduce radiation-induced hematological dysfunction and protect hematopoietic function [76].

### 4.3. Gastrointestinal System

Long-term radiation exposure can cause tissue damage in the gastrointestinal tract, with persistent oxidative stress potentially leading to inflammatory bowel disease and associated malignancies, most notably colitis [77]. One selenoprotein previously associated with colitis is SELENOP. SELENOP acts as an extracellular antioxidant and serves as a selenium transport protein primarily expressed in the liver and secreted into the plasma. Sarah et al. found that SELENOP deficiency increases the risk of inflammatory cancers, emphasizing the critical antioxidant role of epithelial-produced SELENOP in the colonic milieu [78].

### 4.4. Kidney and Liver

Following radiation exposure, the kidney tissue generates a large amount of free radicals, leading to oxidative stress, which can result in renal dysfunction or the onset of kidney disease. Both selenite and SeNPs can protect the kidneys from radiation damage by alleviating oxidative injury. However, the data from Karami et al. showed that SeNPs, an emerging agent for delivering selenium, are more effective than selenite [79]. SeNPs also reduced inflammation and edema in irradiated rats, revealing potential anti-inflammatory activity by inhibiting pro-inflammatory genes and antioxidant effects [80].

Ionizing radiation can also induce hepatotoxicity. Studies have shown that SeNPs may have promising applications in mitigating radiation-induced liver toxicity. NF-κB regulates immunological and inflammatory responses, as well as cell apoptosis. Pharmacological inhibition of NF-κB may be a potential approach for treating radiation injuries. SeNPs may reduce hepatotoxicity by stimulating antioxidant status, increasing Nrf2 expression, enhancing GPXs, and reducing pro-inflammatory marker IL6, lowering NF-κB expression [81]. The discussion by Azmoonfar et al. also demonstrates that SeNPs prevent damage to liver tissues induced by infrared and chemical exposure, confirming their liver-protective action [82].

### 4.5. Cardiovascular System

Cardiovascular diseases account for the majority of global deaths, and radiation-induced cardiovascular disease is one of the main side effects of ionizing radiation exposure. Due to the enormous harm and high mortality caused by cardiovascular damage, it is essential to seek natural and safe therapeutic agents. Amini et al. investigated the combination of selenium-L-methionine and curcumin to determine their capacity to prevent radiation-induced cardiovascular damage. In this study, the combination of selenium-L-methionine and curcumin significantly reduced the expression of pro-inflammatory factors in the heart and blood vessels, indicating its ability to protect cardiovascular health by regulating the redox system and chronic oxidative stress [83]. GPX1 can also repair cardiovascular functional disorders in certain areas. Ye Xiong et al. have found that oxidative stress increases DNA damage, ultimately causing cardiac myocyte cell cycle exit [84]. GPX1 is essential for repairing of cardiovascular damage through DNA repair. Evidence from many other studies indicates that a lack of GPX1 exacerbates heart failure and enhances angiotensin II-mediated cardiac hypertrophy. In contrast, transgenic mice overexpressing GPX1 show higher mitochondrial respiratory efficiency, better contractile strength, and diastolic function compared to wild-type hearts [24,85]. It is evident that GPX1 effectively alleviates cardiovascular damage caused by radiation, protects cardiovascular function, reduces radiation-induced cardiovascular stress, and plays a therapeutic role in radiation treatment.

## 5. Effects of Selenium and Selenoproteins Doses on Radiation Resistance

The evidence for selenium being a chemoprevention agent includes that from animal and epidemiological studies [86,87,88]. However, different results regarding the role of selenium and selenoproteins in chemoprevention have emerged, which may be explained by the selenium status entering the experimental subjects as well as the difference in selenium dose [86]. The reviewed studies and some other ongoing investigations have been providing mounting evidence on the beneficial role of selenium in both healthy people and patients, but it is necessary to pay attention to its status, dose, and type. The dose of selenium and selenoproteins associated with anti-radiation effects may be U-shaped; namely, when the selenium content in the human body is less than ideal situation, selenium and selenium protein may play a beneficial role [89], but when the level of selenium in the human body more than is ideal, excessive amounts of selenium and selenium protein may produce adverse effects [90]. For example, a study conducted by Wang et al. on hens supplemented with three different concentrations of selenium orally for three time periods found that excessive intake of selenium led to decreased immunity and increased oxidative damage [91]. In another study, selenium deficiency caused testicular damage and sperm abnormalities during spermatogenesis [92], and selenium excess impaired sperm quality in male rats [93]. This suggested that non-optimal selenium intake decreased sperm quality and quantity by causing the spermatogenic cells loss and shedding in the testes [94]. Selenium concentration changes will destroy the REDOX state and cause oxidative stress, producing adverse effects.

Although the current experimental results of selenium and selenoproteins as chemopreventive agents are inconsistent, most of these experimental studies support the view that there may be a U-shaped correlation between the doses of selenium and selenoproteins and the anti-radiation effect; that is, the optimal intake of selenium can inhibit oxidative stress, inflammation, apoptosis, etc., and achieve anti-oxidation, DNA repair and immunity enhancement. Therefore, it can be concluded that an optimal Se intake will benefit radiation resistance.

## 6. Summary and Prospects

Selenium is a crucial trace element in the human body, which has great potential for radiation resistance due to its potent antioxidant, DNA repair, and immune regulatory functions. According to current research reports, selenium and selenoproteins enhance antioxidant capacity by clearing hydrogen peroxide and lipid peroxides to combat tissue damage caused by radiation, and exhibit anti-radiation activity through enhancing DNA repair. They protect cells from radiation damage by regulating mitochondrial apoptosis and apoptosis pathways, and also regulate the immune response of the immune system through the MAPK signaling pathway, PI3K/AKT signaling pathway, and inflammatory signaling pathway. Selenium and selenoproteins play an essential role in alleviating and treating radiation damage to organ tissues such as the lungs, kidneys, intestines, cardiovascular systems, etc., with enormous potential.

In the future, we can focus on finding new selenium and selenoproteins with radio-protective functions, and exploring the mechanisms by which different kinds of selenium and selenoproteins work together to protect against radiation. We can also enrich the use of selenium and selenoproteins in multi-organ injury caused by radiation in clinical studies. How to choose the form and dosage of drugs containing selenium will be the challenge. Thus, the complex mechanisms, forms, and dosages of selenium and selenoproteins in anti-radiation activities may become a new research hotspot.

## Figures and Tables

**Figure 1 nutrients-16-02902-f001:**
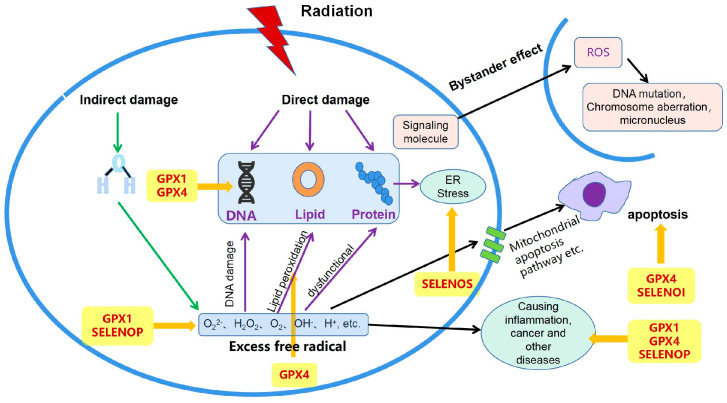
Radiation resistance mechanism of different selenoproteins.

**Table 1 nutrients-16-02902-t001:** Anti-radiation mechanisms of different selenoproteins.

Selenoprotein	Position	Action Mechanism	Reference
GPX1	Erythrocyte	Eliminate reactive oxygen species to reduce radiation damage; Promote DNA repair to prevent radiation injury; Exert anticancer effects to achieve the treat of radiation damage.	[19,20,21,22,23,24,25,26]
GPX4	Cytoplasm, mitochondria and nucleus	Inhibit lipid peroxidation to reduce radiation damage; Regulate apoptotic markers to alleviate radiation-induced cell apoptosis; Decrease the synthesis of 8-oxo-dG to minimize DNA damage.	[27,28,29,30]
SELENOP	Plasma	Inhibition of radiation-induced late reactive oxygen species accumulation; Reduction of tumor occurrence as a tumor suppressor in radiation-induced inflammatory carcinogenesis; Regulation of immune cell function as an antioxidant for immune cells.	[31,32,33,34,35,36,37]
SELENOS	Endoplasmic reticulum	Alleviate radiation-induced endoplasmic reticulum stress.	[38,39]
SELENOI	The Golgi apparatus	Plays a crucial role in T cell activation, enhancing the immune response of immune cells.	[40,41]

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
