# Peer review of "Effect of Selenium and Selenoproteins on Radiation Resistance"

_nutrients, 2024, doi:10.3390/nu16172902_

Round 1
Reviewer 1 Report
Comments and Suggestions for Authors
Zhang et al. submitted a review manuscript on selenium and radiation medicine/DNA damage response, which is novel and interesting. Addressing the following comments shall improve this manuscript.
1. The whole manuscript needs to be carefully proofread not only wordings but also content. For example, the same sentence "Selenium is an essential.......in the body" is shown twice (L16- L19).
2. The emphasis on organic form of selenium and specifically on selenocysteine (L20) in the abstract incompletely covers physiological roles of selenium described in the main text. Strangely, the selenium introduction starts with inorganic selenium compounds (L80) and section 1 does not mention organic selenium at all. Fix.
3. Selenium in excess would otherwise induces oxidative stress, replicative senescence, and DNA damage (Wu et al., Selenium Compounds activate early barriers of tumorigenesis; https://www.jbc.org/article/S0021-9258(19)61086-0/fulltext), which is opposite to the authors unilateral view on the beneficial roles of selenium in radiation resistance. The authors should discuss nutritional/physiological vs. supranutritional/pathophysiological levels of selenium and how selenium contributes to both reduced and oxidative conditions depending on the doses and use the above reference and others to discuss. This avoids the notion of biased and incomprehensive nature of the current coverage of this manuscript.
Comments on the Quality of English LanguageAcceptable
Author Response
- The whole manuscript needs to be carefully proofread not only wordings but also content. For example, the same sentence "Selenium is an essential...in the body" is shown twice (L16-L19).
I am sorry for the mistake of repetition in the abstract. I have cut out the second place and readjust the word order to make the meaning complete.
(L21-L25)
Before: Selenium is an essential trace element for humans and a crucial antioxidant in the body. It can resist radiation by stimulating antioxidant actions, DNA repair functions and immune enhancement. Selenium is an essential trace element for humans and a crucial antioxidant in the body. The human body naturally contains selenium in the forms of selenocysteine (SeCys) and selenoproteins, which are among the many organic forms of selenium that demonstrate low toxicity and high levels of biological activity.
After: Selenium is an essential trace element that functions in the body mainly in the form of selenoproteins. Selenium and selenoproteins can protect against radiation by stimulating antioxidant actions, DNA repair functions and immune enhancement.
- The emphasis on organic form of selenium and specifically on selenocysteine (L20) in theabstract incompletely covers physiological roles of selenium described in the main text. Strangely, the selenium introduction starts with inorganic selenium compounds (L80) and section1 does not mention organic selenium at all. Fix
I appreciate the suggestions by reviewers. My intention was to elicit selenoproteins, so I deleted the part about organic selenium in the abstract and revised it to a statement with selenium and selenoproteins as the main body. This piece of advice has been modified along with the first, and I've integrated them.
(L21-L25)
Before: The human body naturally contains selenium in the forms of selenocysteine (SeCys) and selenoproteins, which are among the many organic forms of selenium that demonstrate low toxicity and high levels of biological activity. Furthermore, selenium and selenoproteins can protect against radiation by stimulating antioxidant actions, DNA repair functions and immune enhancement.
After: Selenium is an essential trace element that functions in the body mainly in the form of selenoproteins. Selenium and selenoproteins can protect against radiation by stimulating antioxidant actions, DNA repair functions and immune enhancement.
- Selenium in excess would otherwise induces oxidative stress, replicative senescence, and DNA damage (Wu et al., Selenium Compounds activate early barriers of tumorigenesis.https://www,jbc.org/article/S0021-9258(19)61086-0/fulltext), which is opposite to the authors unilateral view on the beneficial roles of selenium in radiation resistance. The authors should discuss nutritional/physiological vs. supranutritional/pathophysiological levels of selenium and how selenium contributes to both reduced and oxidative conditions depending on the doses and use the above reference and others to discuss. This avoids the notion of biased and incomprehensive nature of the current coverage of this manuscript.
I am very grateful to the reviewers’ suggestions, which improved the one-sided view of my article. I have added Part 5 of the article as suggested to discuss the effects of dosage regarding selenium and selenoproteins.
After: The evidence for selenium being a chemoprevention agent includes that from animal and epidemiological studies [1-3]. However, different results regarding the role of selenium and selenoproteins in chemoprevention have emerged, which may be explained by the selenium status entering the experimental subjects as well as the difference in selenium dose [1]. The reviewed studies and some other ongoing investigations have been providing mounting evidence on the beneficial role of selenium in both healthy people and patients, but it is necessary to pay attention to its status, dose and type. The dose of selenium and selenoproteins associated with anti-radiation effects may be a U-shaped, namely when the selenium content in the human body is less than ideal situation, selenium and selenium protein may play a beneficial role [4], but when the level of selenium in human body more than is ideal, excessive amounts of selenium and selenium protein may produce adverse effect [5]. For example, a study conducted by Wang et al. on hens supplemented with three different concentrations of selenium orally for three time periods found that excessive intake of selenium leads to decreased immunity and increased oxidative damage [6]. In a study, selenium deficiency caused testicular damage and sperm abnormalities during spermatogenesis [7], and selenium excess impaired sperm quality in male rats [8]. This suggested that non-optimal selenium intake decreased sperm quality and quantity by causing the spermatogenic cells loss and shedding in testis [4]. Selenium concentrations change will destroy the REDOX state and cause oxidative stress, produce adverse effect.
Although the current experimental results of selenium and selenoproteins as chemopreventive agents are inconsistent, most of these experimental studies support the view that there may be a U-shaped correlation between the doses of selenium and selenoproteins and the anti-radiation effect, that is, the optimal intake of selenium can inhibit oxidative stress, inflammation, apoptosis, etc., and achieve anti-oxidation, DNA repair and immunity enhancement. Therefore, it can be concluded that an optimal Se intake will benefit radiation resistance.
- Lippman, S.M.; Klein, E.A.; Goodman, P.J.; Lucia, M.S.; Thompson, I.M.; Ford, L.G.; Parnes, H.L.; Minasian, L.M.; Gaziano, J.M.; Hartline, J.A.; et al. Effect of selenium and vitamin E on risk of prostate cancer and other cancers: the Selenium and Vitamin E Cancer Prevention Trial (SELECT). Jama 2009, 301, 39-51, doi:10.1001/jama.2008.864.
- Shamberger, R.J.; Frost, D.V. Possible protective effect of selenium against human cancer. Can. Med. Assoc. J. 1969, 100, 682.
- Waters, D.J.; Shen, S.; Glickman, L.T.; Cooley, D.M.; Bostwick, D.G.; Qian, J.; Combs, G.F., Jr.; Morris, J.S. Prostate cancer risk and DNA damage: translational significance of selenium supplementation in a canine model. Carcinogenesis 2005, 26, 1256-1262, doi:10.1093/carcin/bgi077.
- Liu, H.; Xu, H.; Huang, K. Selenium in the prevention of atherosclerosis and its underlying mechanisms. Metallomics 2017, 9, 21-37, doi:10.1039/c6mt00195e.
- Wu, M.; Kang, M.M.; Schoene, N.W.; Cheng, W.H. Selenium compounds activate early barriers of tumorigenesis. J. Biol. Chem. 2010, 285, 12055-12062, doi:10.1074/jbc.M109.088781.
- Wang, Y.; Jiang, L.; Li, Y.; Luo, X.; He, J. Effect of Different Selenium Supplementation Levels on Oxidative Stress, Cytokines, and Immunotoxicity in Chicken Thymus. Biol. Trace Elem. Res. 2016, 172, 488-495, doi:10.1007/s12011-015-0598-7.
- Sánchez-Gutiérrez, M.; García-Montalvo, E.A.; Izquierdo-Vega, J.A.; Del Razo, L.M. Effect of dietary selenium deficiency on the in vitro fertilizing ability of mice spermatozoa. Cell Biol. Toxicol. 2008, 24, 321-329, doi:10.1007/s10565-007-9044-8.
- Zhou, J.C.; Zheng, S.; Mo, J.; Liang, X.; Xu, Y.; Zhang, H.; Gong, C.; Liu, X.L.; Lei, X.G. Dietary Selenium Deficiency or Excess Reduces Sperm Quality and Testicular mRNA Abundance of Nuclear Glutathione Peroxidase 4 in Rats. J. Nutr. 2017, 147, 1947-1953, doi:10.3945/jn.117.252544.
Reviewer 2 Report
Comments and Suggestions for Authors
English and scientific language need improvement. For example, the first part of the Introduction is written in a very monotonous manner, often using the same phrases at the beginning of sentences, like: This cases (Line 50), This process, (lines 54, 56), This includes (line 61), These processes (line 62), This result (Line 67) etc.
The sentence in lines 42-43 should be rewritten as it should also reflect selenium and selenoproteins.
Please check all abbreviations and remember that the meaning of an abbreviation should only be used once, the first time it appears. For example, there are many uses for ROS (lines 64, 251, 259 etc.).
Meaning of the sentence on lines 147-149 is not clear: “…reaction between glutathione (GSH) and superoxide, such as hydrogen peroxide,…” What does it mean, for example, “such as hydrogen peroxide”? Please, rephrase.
Figure 1. Typo: Lipin instead Lipid.
Author Response
- English and scientific language need improvement. For example, the first part of the Introductions written in a very monotonous manner, often using the same phrases at the beginning of sentences, like: This causes (Line 50), This process, (lines 54, 56), This includes (line 61), Theseprocesses (line 62), This result (Line 67) etc.
I am very grateful for the reviewers' suggestion, but I am also sorry for my single English expression. I have modified the part of the introduction and changed the expression mode without changing the original meaning.
â‘ (L37-L42)
Before: However, while contributing to industry development, radiation also has some negative effects that cannot be disregarded. These include persisting nuclear leaks from nuclear power plants, potential impacts of radiation from diagnostic and therapeutic equipment, as well as exposure to radiation emissions from electronic products.
After: While promoting industrial development, radiation also has some negative effects that cannot be ignored, such as nuclear leakage, medical radiation exposure, and radiation from electronic products. Recently, Japan's nuclear sewage discharge events increase the risk of radiation to human life, leading to more concerns.
â‘¡(L56-L78)
Before: Direct damage occurs when high-energy ionizing radiation directly affects important macromolecules with biological activity, such as nuclear DNA, proteins or lipid membranes. This causes damage to these molecules and results in pathological changes in body tissues or cell mutations - for example, causing single-strand and double-strand breaks of DNA. Indirect damage is the primary form of ionizing radiation's impact. Ray energy acts on water molecules within cells, causing ionization and excitation. This process produces various reactive oxygen species (ROS) which then attack biological macromolecules like DNA, lipids, proteins, and mitochondria through a free radical chain reaction, resulting in oxidative stress. This process not only induces DNA damage but also inhibits the DNA repair mechanism. Additionally, it causes oxidation of oligosaccharide molecules in the cell membrane to unsaturated free radical dimers, which impairs cell membrane function and leads to protein malfunction. The sustained activation of the DNA damage response also impacts associated signal transduction pathways. This includes the activation of the tumor suppressor p53 signaling pathway and stimulating the expression of cyclin-dependent kinase inhibitors p21 and p16. These processes help maintain age-related growth arrest and induce cell senescence in a coordinated manner. Plenty of reactive oxygen species (ROS) and the evolution of mitochondrial damage may lead to a loss of aerobic respiration complex activity in mitochondria, lower levels of oxidative phosphorylation efficiency, apoptosis, and cascade damage in mitochondria. This results in a significant increase in the overall level of free radicals within cells.
After: Direct damage occurs when high-energy ionizing radiation directly affects important macromolecules with biological activity, such as nuclear DNA, proteins or lipid membranes. The key target of direct damage is DNA, which can result in DNA single-strand and double-strand breaks [9]. Indirect damage is the primary form of ionizing radiation's impact. Ray energy acts on water molecules within cells, causing the ionization of water molecules, producing various reactive oxygen species (ROS), resulting in oxidative stress [10,11]. Under oxidative stress conditions, excessive ROS can cause DNA damage and inhibit DNA repair. Oligosaccharide molecules in the cell membrane are oxidized to unsaturated free radical dimers, which impair the cell membrane function and cause protein dysfunction. Sustained injury response can affect related signal transduction, such as activation of p53 signaling pathway, thereby stimulating the expression of cyclin-dependent kinase inhibitors p21 and p16, causing cell senescence[12]. Mitochondrial damage causes a large amount of ROS to escape, induces the loss of mitochondrial respiratory chain complex activity, reduces the efficiency of oxidative phosphorylation level, causes cell apoptosis, leads to mitochondrial cascade damage, and results in the doubling of free radical level in the whole cell, which continues to cause more serious damage[13].
- The sentence in lines 42-43 should be rewritten as it should also reflect selenium andselenoproteins.
I greatly appreciate the reviewers’ suggestion to make this sentence also relevant to selenium and selenoproteins.
(L42-L47)
Before: It is crucial to study the mechanisms underlying interference caused by radiation damage to establish safe anti-radiation approaches and develop effective anti-radiation drugs.
After: Selenium and selenoproteins have attracted the interest of researchers due to their potential in radiation resistance. Studying the effects and mechanisms of selenium and selenoproteins on radiation resistance will help researchers find safer anti-radiation methods and develop effective anti-radiation drugs.
- Please check all abbreviations and remember that the meaning of an abbreviation should only beused once, the first time it appears. For example, there are many uses for ROS (lines 64. 251.259 etc.).
I apologize for the multiple explanations of ROS in the text. I have checked the acronym list and corrected the remaining errors in the text. ( line 147.174.186 etc.)
- Meaning of the sentence on lines 147-149 is not clear:"...reaction between glutathione (GSH)and superoxide, such as hydrogen peroxide,..." What does it mean, for example, “such as hydrogen peroxide"? Please. Rephrase.
I am sorry that the expression of this sentence is not clear, and I have revised it according to the suggestion.
(L183-L185)
Before: GPX1 can catalyze the reaction between glutathione (GSH) and superoxide, such as hydrogen peroxide, to regulate the accumulation of hydrogen peroxide.
After: GPX1 can catalyze the reaction between glutathione (GSH) and some superoxide, to regulate the accumulation of these superoxide.
- Figure 1. Typo: Lipin instead Lipid.
I am sorry for the input error in the picture, which I have corrected.
- Havránková, R. Biological effects of ionizing radiation. Cas Lek Cesk 2020, 159, 258-260.
- Meador, J.A.; Morris, R.J.; Balajee, A.S. Ionizing Radiation-Induced DNA Damage Response in Primary Melanocytes and Keratinocytes of Human Skin. Cytogenet. Genome Res. 2022, 162, 188-200, doi:10.1159/000527037.
- Carante, M.P.; Ramos, R.L.; Ballarini, F. Radiation Damage in Biomolecules and Cells 2.0. Int J Mol Sci 2023, 24, doi:10.3390/ijms24043238.
- Ahmed, K.M.; Li, J.J. NF-kappa B-mediated adaptive resistance to ionizing radiation. Free Radic Biol Med 2008, 44, 1-13, doi:10.1016/j.freeradbiomed.2007.09.022.
- Liu, L.; Liang, Z.; Ma, S.; Li, L.; Liu, X. Radioprotective countermeasures for radiation injury (Review). Molecular Medicine Reports 2023, 27, doi:10.3892/mmr.2023.12953.